# Effects of Zearalenone on Apoptosis and Copper Accumulation of Goat Granulosa Cells In Vitro

**DOI:** 10.3390/biology12010100

**Published:** 2023-01-09

**Authors:** Liang Liu, Jianyu Ma, Zongyou Wei, Yingnan Yang, Dongxu Li, Yongjie Wan

**Affiliations:** 1Jiangsu Livestock Embryo Engineering Laboratory, College of Animal Science and Technology, Nanjing Agricultural University, Nanjing 210095, China; 2Enterprise Graduate Workstation, Taicang Agricultural and Rural Science & Technology Service Center, Taicang 215400, China

**Keywords:** goat granulosa cell, zearalenone, cytotoxicity, copper transport

## Abstract

**Simple Summary:**

Zearalenone (ZEA) is a prominent feed contaminant worldwide, causing considerable financial losses in the livestock industry due to its estrogen-like effect. Although many reports have claimed that ZEA can interfere with the reproductive system of mammals, little confirmatory information is available. We used ZEA to treat goat granulosa cells (GCs) and found that ZEA could promote apoptosis and oxidative stress levels in GCs, inhibit cell proliferation and arrest the cell cycle. RNA-Seq was performed, which also showed that copper transport-related gene expression was altered, leading to copper accumulation in GCs. These findings provide a new perspective for research on the relationship between ZEA toxicity and copper accumulation

**Abstract:**

Zearalenone (ZEA), also known as F-2 toxin, is a mycotoxin. Despite numerous reports of ZEA impairing livestock production performance and fertility, little information is available, including information about the mechanism underlying damage to cell metal ion transport. Copper, which is essential for cell survival as a metal ion, can consist of a variety of enzymes that facilitate abundant metabolic processes. However, the accumulation of copper in cells can have toxic effects. Here, we intended to determine whether ZEA could impair goat granulosa cells (GCs) and alter the cellular copper concentration. GCs were divided into a negative control (NC) group (cells cultured with 0.1% dimethyl sulfoxide (DMSO) for 8 h) and a ZEA group (cells cultured with 200 μmol/L ZEA diluted in DMSO for 8 h). The results showed that ZEA could inhibit GC proliferation and impair cell viability. GCs showed significant increases in the apoptosis rate and oxidative stress levels, while their ability to synthesize estrogen decreased. In addition, RNA-seq results showed dramatic changes in the expression of copper transport-related genes. The expression levels of ATPase copper transporting alpha (*ATP7A*) and ATPase copper transporting beta (*ATP7B*) were significantly downregulated (*p* < 0.01), while the expression of solute carrier family 31 member 1 (*SLC31A1*) was not modified in the ZEA group compared with the NC group. In accordance with these trends, the copper concentration increased significantly in the ZEA group (*p* < 0.01). In summary, our results show that ZEA can negatively affect GCs and cause copper accumulation. This finding may provide a prospective line of research on the relationship between ZEA and the transport of copper ions in GCs.

## 1. Introduction

It is well known that the safety of food and byproducts is of constant concern. Mycotoxins, due to their complex chemical properties, are difficult to eliminate entirely [1]. Leftover mycotoxins in feed ingredients or food can cause a range of diseases, including endocrine dysplasia and reproductive dysfunction, among others [2]. ZEA, also known as F-2 toxin, is a mycotoxin that spreads worldwide. Previous studies have shown that ZEA can cause blood and liver toxicity [3], immunotoxicity [4], carcinogenicity [5], mutagenicity [6], and genotoxicity [7]. Even at extremely low concentrations, ZEA contamination may cause fetal death or neonatal weight loss during pregnancy in mammals [8]. Because of its estrogen-like effect, ZEA can inhibit ovarian activity and subsequently impair the fertility of female animals [9].

GCs are particularly vital for the development of the female reproductive system. Studies have shown that atresia of the follicle is associated with apoptosis of granulosa cells [10]. GCs also secrete estrogen, which is indispensable for follicle growth, depending on whether the follicle would otherwise be blocked or develop to the next stage [11]. However, there are abundant estrogen receptors located in the GC membrane [12]. Therefore, when ZEA is taken in, GCs are most likely to be affected [13]. Eventually, the whole female reproductive system will be adversely influenced.

Copper is essential for cell survival and can consist of various enzymes that participate in numerous metabolic processes, such as the tricarboxylic acid (TCA) cycle [14] and mitochondrial respiratory chain [15]. However, excessive accumulation of copper can have a toxic effect [16]. The excessive amount of copper in the cell weakens the stability of the Fe-S protein cluster and interrupts the TCA cycle process, leading to metabolic intermediate accumulation and protein misfolding. However, the relationship between ZEA exposure and copper accumulation in GCs has rarely been studied. Therefore, we wondered if ZEA might cause granulosa cell death through copper accumulation.

## 2. Materials and Methods

All experiments were conducted according to the guidelines of the Institutional Ani mal Care and Use Committee of Nanjing Agricultural University.

### 2.1. Cell Culture

Goat GCs were collected from follicles in the ovaries, which were obtained from Danyang abattoir (Danyang, Jiangsu, China) within 4 h after slaughter. Cells were cultured in 6-well culture plates supplemented with complete medium consisting of DMEM/F12 glutaMAX (Gibco, Thermo Fisher Scientific, Waltham, MA, USA), 10% fetal bovine serum (FBS) and 100 U/mL penicillin/streptomycin (P/S). All cells were incubated in a 5% CO_2_ atmosphere at 37 °C. The complete medium was changed every 24 h.

### 2.2. Cell Counting Kit—8 Assays

Cells were seeded in 96-well culture plates with complete medium containing specified concentrations of ZEA (0, 50, 100 and 200 μmol/L), which was diluted in dimethyl sulfoxide (DMSO). After treatment with ZEA for 8 h, 10 µL Cell Counting Kit-8 (CCK-8) reagent was added to each well and incubated (MedChemExpress, Monmouth Junction, NJ, USA) for approximately 4 h at 37 °C and 5% CO_2_. The absorbance at 450 nm was detected by a spectrophotometer (Thermo Fisher Scientific, Waltham, MA, USA). According to the CCK-8 results (Appendix A), the concentration of 200 μmol/L ZEA was chosen as the ZEA group, and 0 μmol/L zearalenone was chosen as the NC group, the CCK-8 result is represented in Appendix A.

### 2.3. Cell Proliferation Analysis

Cell proliferation was assessed by the EdU Assay Kit (KGA332-50, Beyotime Biotechnology, Nantong, China). GCs were seeded in a 24-well culture plate with a circular cap at the bottom. When the cells attached to the coverslip, they were first treated with 0 and 200 μmol/L zearalenone for 8 h. The complete medium with ZEA was then replaced by complete medium containing 100 mmol/L EdU. After 2 h of incubation at 37 °C and 5% CO_2_, the cells were washed with DPBS 3 times and fixed with 4% paraformaldehyde (PFA) for 30 min at RT. Next, the cells were washed with 2 mg/mL glycine for 5 min and permeabilized with DPBS containing 0.5% Triton X-100 for approximately 20 min. DPBS was used to wash the cells again. Subsequently, the cell nuclei were stained with Hoechst 33342 for 5 min at RT. Finally, the samples were analyzed by confocal laser scanning microscopy (Carl Zeiss, Oberkochen, Germany).

### 2.4. RNA Extraction and Quantitative Real-Time PCR (qRT-PCR)

Total RNA from granulosa cells was extracted using TRIzol reagent (Invitrogen, Shanghai, China). Then, cDNA was obtained using reverse transcription reagent kits (Vazyme Biotech Co., Ltd., Nanjing, China) according to the manufacturer’s instructions. All of the quantitative real–time PCR procedures were performed in QuantStudio™ Design & Analysis Software (Applied Biosystems, Foster Cite, CA, USA) with Chamq Universal SYBR qPCR Master Mix (Vazyme Biotech Co., Ltd., Nanjing, China). All primers used for qRT-PCR are listed in Appendix A.

### 2.5. Protein Extraction and Western Blot Analysis

After treatment with the specified concentrations of ZEA for 8 h, cells were washed with DPBS and lysed with RIPA buffer containing protease inhibitors (Thermo Pierce, Rockford, IL, USA) and phosphatase inhibitors. After approximately 20 min, an ultrasound cell disruptor was used to ensure that the cells were completely disrupted. Next, the protein concentrations were assessed using a BCA protein assay kit (Beyotime, Nanjing, China). Equal amounts of protein samples were subjected to 12% SDS–polyacrylamide gel electrophoresis (SDS-PAGE) and transferred to polyvinylidene fluoride (PVDF) membranes (Millipore; Billerica, MA, USA). Then, 5% BSA in Tris–buffered saline–Tween (TBST) was used to block the membranes for 2 h. The membranes were incubated overnight at 4 °C with different primary antibodies. Next, the membranes were washed with TBST for 2 h and incubated with secondary antibodies for another 1 h. Subsequently, protein bands were measured using enhanced chemiluminescence (ECL) reagent (Thermo Pierce, USA) and exposed with Image Quant LAS 400 (Fiji Film, Tokyo, Japan). The results were analyzed with ImageJ software (Wayne Rasband, MD, USA). 

### 2.6. Flow Cytometry Analysis of the Cell Cycle and Apoptosis

Cells treated with a specified concentration of ZEA for 8 h were harvested. For cell cycle analysis, cells were collected and fixed in 70% ethanol at −20 °C overnight. The samples were then sent to Wuhan Servicebio Ltd. for testing. For cell apoptosis analysis, all cells, including those suspended in complete medium, were harvested and stored in DPBS. The samples were sent to Nanjing Liangwei Biotechnology Co., Ltd. (Nanjing, China) for flow cytometry analysis within 2 h.

### 2.7. Atomic Absorption Spectrum (AAS) Measurement of Copper

Cells were harvested after being treated with ZEA for 8 h. After washing cells with PBS 3 times, 200 μL ultrapure water was used to resuspend cells. Then, the samples were added to the tube along with 10 mL of pure nitric acid and were digested for approximately 1.5 h. Next, ultrapure water and nitric acid were evaporated at 160 °C for 4 to 5 h until there was no liquid, and 10 mL ultrapure water was used to dilute the dry matter and measure the copper concentrations of different samples by atomic absorption spectroscopy.

### 2.8. Statistical Analysis

Data were analyzed using SPSS 24.0 software (SPSS Inc., Chicago, IL, USA). The significance of the differences between samples was compared using a *t*-test for two groups and a one-way ANOVA for three or more groups. Each experiment was repeated at least three times. The data are expressed as the mean value ± standard error (SEM). *p* < 0.05 (*) and *p* < 0.01 (**) indicate significant and extremely significant differences, respectively.

## 3. Results

### 3.1. ZEA Treatment Decreased Proliferation and Cell Viability in GCs

ZEA was adopted to examine whether it would restrain the proliferation of goat GCs. After treatment with ZEA for 8 h, compared with the NC group, there was a highly significant decrease in EdU-positive cell incorporation in the ZEA group (*p* < 0.01) (Figure 1A). Cell viability was measured using the CCK-8 kit, which also showed a severe decrease in the ZEA group (*p* < 0.01) (Figure 1B). Expression of the cell proliferation-related gene PCNA was detected at both mRNA and protein levels, and the mRNA expression of PCNA was extremely significantly reduced (*p* < 0.01) (Figure 1C), while the protein level of PCNA was significantly reduced (*p* < 0.05) (Figure 1C) in the ZEA group. The above results indicated that ZEA reduced the viability of GCs and inhibited GC proliferation.

### 3.2. Cell Cycle Arrest with ZEA Treatment

To further explore the potential negative effects of ZEA on cell proliferation efficiency, a flow cytometry analysis of the cell cycle was also performed. The results indicated that the cell cycle was arrested at the G1/S phase, while there was a significant increase (*p* < 0.05) in the G1 fraction and a highly significant decrease (*p* < 0.01) in the S phase (Figure 2A). In addition, the expression of genes related to the G1/S phase, such as CCND1, CDK4 and CDK6, was measured at both mRNA and protein levels. The expression levels of all genes, including CCND1, CDK4 and CDK6, were significantly decreased (*p* < 0.01) (Figure 2B). At the protein level, CDK4 expression was significantly reduced (*p* < 0.01), and CCND1 and CDK6 were significantly reduced (*p* < 0.05) (Figure 2C). These results demonstrated that ZEA treatment did have a negative effect on GC proliferation, which would impact the cell proliferation efficiency by arresting the cell cycle at the G1/S phase.

### 3.3. ZEA Treatment Promoted Cell Apoptosis

Flow cytometry was performed to analyze cell apoptosis. In comparison to the NC group, the percentage of apoptotic cells was extremely significantly elevated (*p* < 0.01) in the ZEA group (Figure 3A). Additionally, apoptosis-related genes, such as BCL2-associated X, apoptosis regulator (BAX), Caspase 3, Caspase 9 and tumor protein p53 (p53), were altered. The mRNA expression levels of BAX, Caspase 3, and Caspase 9 were very significantly promoted (*p* < 0.01) (Figure 3B), while the increase in p53 was not significant (Figure 3B). The anti-apoptotic gene BCL2 apoptosis regulator (Bcl-2) expression level and the ratio of Bcl-2/BAX at the mRNA level were significantly decreased (*p* < 0.01) (Figure 3B). The protein expression levels of BAX, Caspase 3, Caspase 9 and p53 were highly significantly increased (*p* < 0.01) (Figure 3C,D), while the expression level of Bcl-2 was significantly decreased (*p* < 0.05) (Figure 3C,D) in the ZEA group. Overall, ZEA exposure improved the apoptosis process in goat GCs.

### 3.4. ZEA Treatment Triggered Oxidative Stress and Mitochondrial Dysfunction in GCs

It has been demonstrated that ROS accumulation induced by mycotoxins can impair cell viability and inhibit cell proliferation [17]. Therefore, we detected ROS levels and oxidative stress-related genes, including superoxide dismutase 2 (*SOD2*), catalase (*CAT*) and *GSH-px*. The results showed that the ROS level (Figure 4A) and mRNA expression levels of *SOD2*, *CAT* and *GSH-px* increased very significantly (*p* < 0.01) (Figure 4B) when GCs were treated with ZEA. The protein expression levels of SOD2 and CAT also improved quite significantly (*p* < 0.01) (Figure 4C), which meant that cells were adjusted to the steep ROS circumstance by promoting antioxidative gene expression. We also detected the mitochondrial membrane potential (MMP) by JC-1 staining. The JC-1 (Red)/JC-1 (Green) ratio decreased extremely significantly (*p* < 0.01) (Figure 4D), indicating that MMP was affected by ZEA. A lower MMP also indicated damaged mitochondria. The above evidence suggested that ZEA caused ROS accumulation and triggered oxidative stress in GCs.

### 3.5. ZEA Treatment Disturbed the Estrogen Synthesis in GCs

According to the above-mentioned results, we wondered if ZEA would also have a negative effect on estrogen synthesis, which is a major function of GCs. To explore our hypothesis, estrogen synthesis-related gene expression was detected. The 3beta-hydroxysteroid dehydrogenase/isomerase (*3B-HSD*), cytochrome P450 family 11 subfamily A member 1 (*CYP11A1*) and cytochrome P450 family 17 subfamily A member 1A (*CYP17A1*) mRNA expression levels decreased significantly (*p* < 0.05), while cytochrome P450 family 19 subfamily A member 1 (*CYP19A1*) decreased with no significance in the ZEA group compared with the NC group (Figure 5A). The mRNA expression of steroidogenic acute regulatory protein (*StAR*) increased significantly (*p* < 0.05) (Figure 5A). Additionally, the protein expression level of CYP11A1 decreased extremely significantly (*p* < 0.01), while the decrease in CYP19A1 was not significant (Figure 5B), in agreement with the higher concentration of progesterone (*p* < 0.01) and lower concentration of estrogen (*p* < 0.01) in the ZEA group (Figure 5C). These results illustrated that ZEA had a negative influence on estrogen synthesis in GCs.

### 3.6. ZEA Treatment Increased Copper Accumulation in GCs

Based on our previous results, we further investigated whether ZEA affected different cell death pathways, such as the transport of various metal ions, in addition to its effects on classical cell death pathways. Next, we performed transcriptome sequencing. Based on the RNA-Seq results, we found that copper transport-related gene expression was dramatically changed (Figure 6A–D). The mRNA expression level of metallothionein-2 (*MT2A*), which could reflect the copper level in cells, increased significantly (*p* < 0.01) (Figure 6C,D), indicating the potential for copper accumulation in GCs after ZEA treatment. We then examined the expression of the copper transporters *ATP7A* and *ATP7B*. Both *ATP7A* and *ATP7B* mRNA expression levels decreased extremely significantly (*p* < 0.01) (Figure 6D), and ATP7A protein levels decreased significantly (*p* < 0.05), while the reduction in ATP7B was not significant (Figure 6E). The expression of the copper importer SLC31A1 was not altered at the protein level (Figure 6E). Therefore, we detected the intracellular copper levels and demonstrated that copper accumulation occurred after ZEA exposure. The concentration of copper in the ZEA group was twice as high as that in the NC group (*p* < 0.01) (Figure 6F).

### 3.7. ZEA Treatment Interfered with the TCA Cycle Related Enzyme Gene Expression and Rose Protein Toxic Stress in GCs

After confirming that ZEA could cause copper accumulation in GCs, we examined TCA cycle–related gene expression. Previous studies have demonstrated that excessive amounts of copper in cells can interfere with the TCA cycle, leading to the accumulation of metabolic intermediates that can trigger protein misfolding [14]. The accumulation of misfolded proteins eventually results in cell death [14]. qRT-PCR was performed to detect the expression of lipoyltransferase 1 (*LIPT1*), lipoic acid synthetase (*LIAS*), dihydrolipoamide dehydrogenase (*DLD*), dihydrolipoamide S-acetyltransferase (*DLAT*), pyruvate dehydrogenase E1 subunit alpha 1 (*PDHA1*) and pyruvate dehydrogenase E1 subunit beta (*PDHB*). Consistent with the RNA-Seq results (Figure 7A–C), all genes mentioned above had significantly reduced expression levels of mRNA (*p* < 0.01) (Figure 7D). In addition, the expression of heat shock protein 70.1 (*HSP70*) increased very significantly at both the mRNA and protein levels (*p* < 0.01) (Figure 7E,F). All the evidence suggested that the TCA cycle was partly blocked and the protein toxic stress level rose in GCs in response to ZEA treatment.

## 4. Discussion

ZEA has received widespread attention because of the estrogen-like toxic effects of its metabolites [18]. Due to its stable chemical structure and strong thermal stability, it is difficult to completely remove it from feed ingredients, and improper storage of feed can still breed fresh ZEA toxins [19]. Previous studies have confirmed that ZEA reduces fecundity in many livestock [20,21,22]. However, its specific mechanism has not been thoroughly studied.

In our present experiment, ZEA showed its ability to suppress the proliferation of goat GCs, arresting most of the cell cycle in the G1/S phase transition state. Our results were consistent with the results of Yi, Y., et al. [23], who reported that ZEA treatment could lead to cell cycle arrest at the G1/S phase. However, the results of Li, N., et al. [24] demonstrated that ZEA treatment arrested the cell cycle of porcine GCs in G2/M phase. The differences between the two reports may arise from the different treatment doses and timing of ZEA. The cell viability of granulosa cells also decreased significantly after ZEA treatment to only 60–70% of normal cell levels, in accordance with the results of Chen, F., et al. [25], indicating that GC cell viability was significantly inhibited by ZEA at 30–150 μmol/L in a dose-dependent manner. Overall, these results illustrate that ZEA treatment can arrest the GC cell cycle and in turn adversely affect GC proliferation.

GC proliferation and apoptosis are intimately associated with the development of ovarian follicles [26]. The increased apoptosis rate of GCs not only directly reduces the number of GCs but also weakens their ability to synthesize steroid hormones [22]. Zhu, L., et al. [27] claimed that ZEA could induce GC apoptosis by activating the caspase 3- and caspase 9-dependent mitochondrial signaling pathways, which was confirmed in our study. In addition, Liu, X.L., et al. [6] demonstrated that ZEA exposure impaired the genomic stability of swine follicular granulosa cells in vitro. Oxidative stress is also a main cause of the ability of ZEA to trigger apoptotic processes in GCs. A previous study elucidated that elevated ROS can impair mammalian reproductive cells [28], as demonstrated in our results. Synthesizing and secreting steroid hormones are the main functions of GCs. Nevertheless, ZEA exposure can interfere with the synthesis of estrogen by interrupting the expression of *3BHSD* [29]. Our findings complement the results, showing the negative impact of ZEA on the expression of *CYP11A1* and *CYP19A1*. Altogether, these results demonstrate that ZEA promotes apoptosis in GCs, causing oxidative stress and subsequently impairing estrogen synthesis.

In addition to the above-mentioned classical cell death pathway, we wondered if ZEA could alter other aspects that are essential for cell survival, such as the transport of metal ions. Evidence has shown that ZEA can increase the Fe^2+^ and Fe^3+^ concentration on mice spermatogenesis, leading to a higher ferroptosis level [30]. T–2 toxin promoted ferroptosis by inducing lipid ROS and decreased the expression of solute carrier family 7 member 11 (SLC7A11) [31]. Most metallic elements are in the form of metal ions in cells [32]. They can produce many bonds and cooperate with biomolecules, thus playing a variety of roles in organisms, such as electron transport [33], gas transport [34], and enzyme activity centers [35]. Following the RNA sequencing results, we found significant changes in genes and pathways associated with metal ion transport, particularly copper ion transport. The level of copper ions in the cell is primarily determined by the copper ion transfer genes *SCL31A1* and *ATP7A*/*ATP7B* [36]. *SCL31A1* transports copper ions into the cell, facilitating their binding to various metabolic enzymes to perform their catalytic functions [37]. *ATP7A*/*ATP7B* removes excess copper ions from cells into body fluids, where most of them are eventually excreted via liver and bile metabolism [38]. During the experiment, we observed that treatment with ZEA resulted in an increase in the apoptosis rate and the disruption of normal physiological function. Moreover, the sequencing results showed a significant decrease in expression of the *ATP7A*/*ATP7B* genes, while there was no difference in the expression levels of the *SCL31A1* gene. Subsequent qRT-PCR and WB experiments also confirmed that the expression trends of these genes were consistent with the sequencing results. Using AAS to detect copper levels before and after ZEA treatment, it was found that the ZEA group had approximately twice the copper levels of the NC group. The expression level of *HSP70* also increased in the ZEA group, similar to the results of the new copper death model [39]. Therefore, we hypothesize that granulosa cell damage due to ZEA treatment may be associated with intracellular copper accumulation. ZEA causes the accumulation of copper in cells by reducing the expression of *ATP7A*/*ATP7B*. Excessive copper ultimately increases the levels of proteotoxic stress, leading to cell death [40]. 

In summary, we first found that ZEA can lead to copper accumulation in GCs and provoke proteotoxic stress. This finding may provide a new perspective, and related research is needed in the future.

## 5. Conclusions

In conclusion, we found that ZEA can negatively regulate the proliferation of goat GCs and promote apoptosis. In addition, we are the first group to shed light on the idea that ZEA can lead to copper accumulation in GCs. This finding may provide a fresh perspective for research on the relationship between ZEA cytotoxicity and copper transport in cells.

## Figures and Tables

**Figure 1 biology-12-00100-f001:**
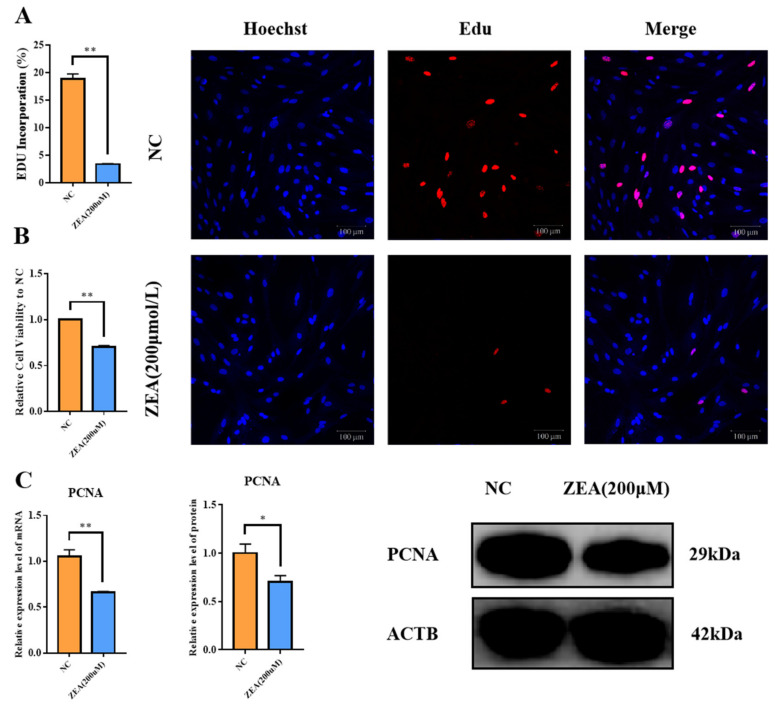
Effect of ZEA treatment on viability and proliferation of GCs. (**A**): EdU positive cell incorporation of the NC and the ZEA groups. (**B**): Cell viability of the NC and the ZEA groups. (**C**): The expression of PCNA in mRNA and protein levels of the NC and the ZEA groups. All the experiments were performed in triplicate, each value is the mean ± SEM. * *p* < 0.05, ** *p* < 0.01.

**Figure 2 biology-12-00100-f002:**
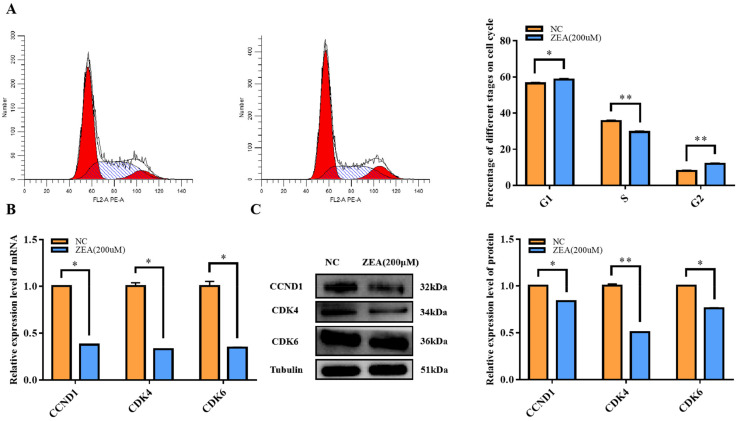
Effect of ZEA treatment on GCs cycle. (**A**): Changes in cell cycle after GCs were treated with zearalenone (200μmol/L) for 8 h. (**B**): The expression of cell cycle related genes *CCND1*, *CDK4* and *CDK6* in mRNA level of the NC and the ZEA groups. (**C**): The expression of cell cycle related genes CCND1, CDK4 and CDK6 in protein levels of the NC and the ZEA groups. All the experiments were performed in triplicate, each value is the mean ± SEM. * *p* < 0.05, ** *p* < 0.01.

**Figure 3 biology-12-00100-f003:**
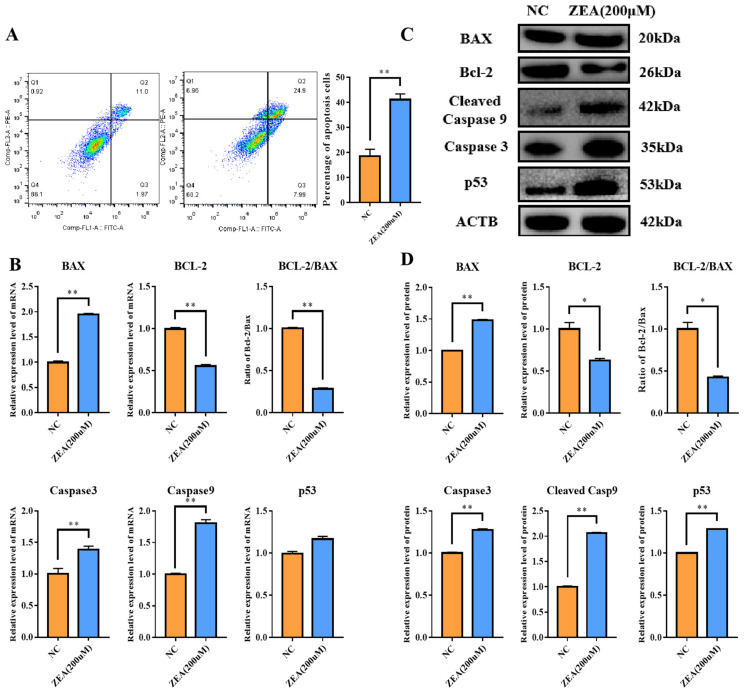
Effect of ZEA treatment on apoptosis in GCs. (**A**): The apoptotic rate of GCs detected by flow cytometry. (**B**): Expression levels of *BAX*, *Caspase3*, *Caspase9*, *p53* and *Bcl-2* in mRNA in the NC and the ZEA groups detected by qRT-PCR. (**C**,**D**): Expression levels of BAX, Caspase3, Caspase9, p53 and Bcl-2 in the NC and the ZEA groups in protein detected by Western blot. All the experiments were performed in triplicate, each value is the mean ± SEM. * *p* < 0.05, ** *p* < 0.01.

**Figure 4 biology-12-00100-f004:**
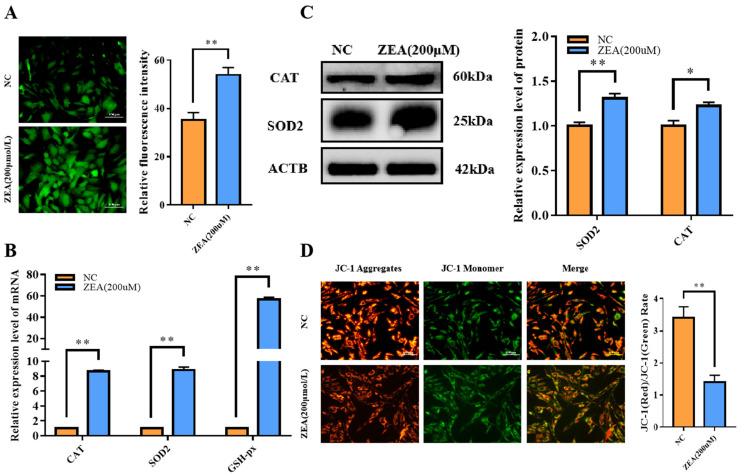
Effect of ZEA treatment on oxidative stress in GCs. (**A**): ROS level of the NC and the ZEA groups. (**B**): The expressions of oxidative stress-related genes including *SOD2*, *CAT* and *GSH-px* in mRNA level of the NC and the ZEA groups. (**C**): The expression of SOD2 and CAT in protein level of the NC and the ZEA groups. (**D**): JC-1 staining result of the NC and the ZEA groups. All the experiments were performed in triplicate, each value is the mean ± SEM. * *p* < 0.05, ** *p* < 0.01.

**Figure 5 biology-12-00100-f005:**
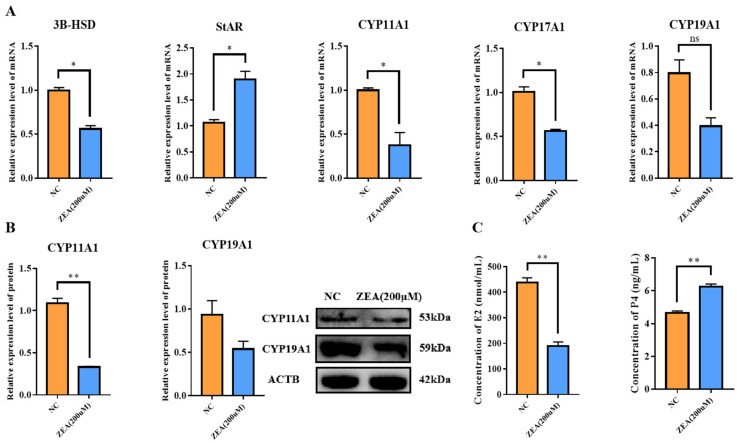
The effects of ZEA treatment on GCs estrogen synthesis. (**A**): The mRNA expression levels of *3B-HSD*, *StAR*, *CYP11A1*, *CYP17A1* and *CYP19A1* measured by qRT-PCR in the NC and the ZEA groups. (**B**): The protein expression level of CYP11A1 and CYP19A1 measured by Western blot in the NC and the ZEA groups. (**C**): The concentration of estrogen and progesterone in the NC and the ZEA groups, measured by E2 and P4 ELISA Kit. All the experiments were performed in triplicate, each value is the mean ± SEM. * *p* < 0.05, ** *p* < 0.01, ns, not significant.

**Figure 6 biology-12-00100-f006:**
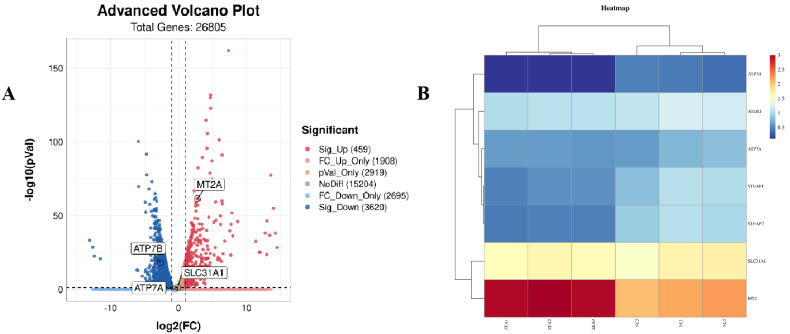
Zearalenone treatment caused copper accumulation in GCs. (**A**): Volcano plot of copper transfer related genes including SLC31A1, MT2A, ATP7A and ATP7B. (**B**): Heatmap of expression level of copper transfer genes. (**C**): Log10 of FPKM of copper transfer genes. (**D**): mRNA expression levels of copper transfer genes. (**E**): protein expression levels of ATP7A and SCL31A1 in the NC and the ZEA groups. (**F**): Intracellular copper concentration of the NC and the ZEA groups. All the experiments were performed in triplicate, each value is the mean ± SEM. * *p* < 0.05, ** *p* < 0.01.

**Figure 7 biology-12-00100-f007:**
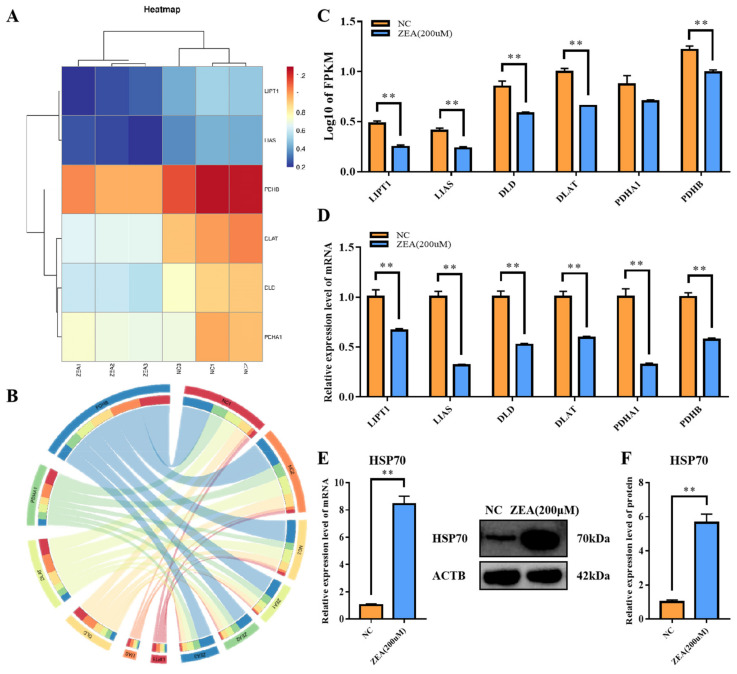
ZEA treatment interfered with the TCA cycle related enzyme gene expression and increased protein toxic stress in GCs. (**A**): Heatmap of TCA cycle related genes including *LIPT1*, *LIAS*, *DLD*, *DLAT*, *PDHA1* and *PDHB*. (**B**): Circos of expression level about TCA cycle related genes. (**C**): FPKM of TCA cycle related genes. (**D**): mRNA expression levels of TCA cycle related genes. (**E**): mRNA expression level of *HSP70* in the NC and the ZEA groups. (**F**): protein expression level of HSP70 in the NC and the ZEA groups. All the experiments were performed in triplicate, each value is the mean ± SEM. ** *p* < 0.01.

## Data Availability

Not applicable.

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
