# Peer review of "Effects of Zearalenone on Apoptosis and Copper Accumulation of Goat Granulosa Cells In Vitro"

_biology, 2023, doi:10.3390/biology12010100_

Round 1

Reviewer 1 Report (Previous Reviewer 1)

After the author's revision, the manuscript has reached the level of publication, and it is recommended to accept

Author Response

Thank you very much for your affirmation of the article, which gives us great encouragement. Also, we really appreciate  your suggestions before, which have helped us improve the quality of the article.

Reviewer 2 Report (Previous Reviewer 2)

Dear, despite the effort which has been spent by authors to improve the manuscript, doubt about the significance and importance of research is still present. The current statements in the Conclusion are the opposite of than in the previous version of the manuscript, making this susceptible.
Likewise, do authors have experience in such kind of research? This research has not been supported by self-cited references. Statements in the section Discussion, conclusions, and manuscript as a whole are not supported by the listed self-citations which did not confirm previous research on this topic.
The article has serious flaws related to the physiology mechanism of zearalenone and copper. Copper belongs to the essential microelements with many vital functions in organisms. Therefore, the research was not conducted correctly.

Author Response

Response to Reviewer 2 Comments

Dear, despite the effort which has been spent by authors to improve the manuscript, doubt about the significance and importance of research is still present. The current statements in the Conclusion are the opposite of than in the previous version of the manuscript, making this susceptible.
Likewise, do authors have experience in such kind of research? This research has not been supported by self-cited references. Statements in the section Discussion, conclusions, and manuscript as a whole are not supported by the listed self-citations which did not confirm previous research on this topic.
The article has serious flaws related to the physiology mechanism of zearalenone and copper. Copper belongs to the essential microelements with many vital functions in organisms. Therefore, the research was not conducted correctly.

  1. The current statements in the Conclusion are the opposite of than in the previous version of the manuscript, making this susceptible.

Response 1: Thank you very much for your comments. First of all, we deeply regret the misunderstanding caused by our presentation. During our experiments, we found that ZEA were able to alter the expression levels of copper transport proteins, including SCL31A1, ATP7A and ATP7B. As a result, the intracellular copper concentration increased considerably and copper accumulation occurred. In addition, we observed a decrease in the TCA cycle associated with the expression of key genes, indicating that the TCA cycle was also impaired. The results were similar to those of a newly discovered pathway of cell death called cuprotosis. So, we conjecture that cuprotosis may be involved in the ZEA exposure process. However, after reading your comments for Round 1, we did find it was inappropriate to claim that cuprotosis was related to ZEA treatment when we were not able to prove it directly. After referring to your valuable advice, we changed our statement to focus only on the accumulation of copper in the cell, which can be directly demonstrated by our experimental results. More importantly, the relationship between the accumulation of copper in the cell and cytotoxicity of ZEA has not been studied before, so we consider this as a fresh research perspective. Hope our answers can satisfy you.

  1. This research has not been supported by self-cited references. Statements in the section Discussion, conclusions, and manuscript as a whole are not supported by the listed self-citations which did not confirm previous research on this topic.

Response 2: Thank you so much for your comments. Since copper death is a novel cell death mechanism that has only recently been discovered, the correlation between copper death and ZEA cytotoxicity has not been studied. However, some papers aiming to study the relationship between mycotoxin and ferroptosis have proved that ZEA treatment can increase the Fe2+ and Fe3+ concentration on mice spermatogenesis, leading to a higher ferroptosis level [1]. In addition, T-2 toxin promoted ferroptosis by inducing lipid reactive oxygen species (ROS), as N-acetyl-l-cysteine significantly blocked T-2 toxin-induced ferroptosis. Moreover, T-2 toxin decreased the expression of solute carrier family 7 member 11 (SLC7A11) and failed to further enhance ferroptosis in SLC7A11-deficient cells [2]. Those results mentioned above are similar to our results that ZEA can alter the copper transport protein expressions and cause copper accumulation. What we can learn now is that copper will be involved in a variety of cellular metabolic processes, including the synthesis of the SOD family of proteins, which are essential for dealing with oxidative stress. Exposure to ZEA induces a disorder in the levels of oxidative stress in the cell, and thus the synthesis of the SOD protein family is affected. Therefore, we infer that the copper levels in the cell will change accordingly, which is also proved by our results. The corresponding literature support is also listed in the discussion section. Hope our answer can satisfy you.

  1. Li, Y., et al., Effect of Zearalenone-Induced Ferroptosis on Mice Spermatogenesis. Animals (Basel), 2022. 12(21).
  2. Wang, G., et al., T-2 Toxin Induces Ferroptosis by Increasing Lipid Reactive Oxygen Species (ROS) and Downregulating Solute Carrier Family 7 Member 11 (SLC7A11). J Agric Food Chem, 2021. 69(51): p. 15716-15727.

Reviewer 3 Report (New Reviewer)

The paper titled Effects of zearalenone on apoptosis and copper accumulation of goat granulosa cells in vitro is an interesting study aimed at determining whether the mycotoxin zearalenone (ZEA) can cause granulosa cell death by copper accumulation
ZEA is a known mycotoxin because it is widely distributed. The most sensitive species are pigs. The author chose granulosa cells and I miss an explanation why. Also, please explain the used concentration of 200 ug from the point of view of daily dose intake. Is such a concentration realistic or is it just for scientific research to investigate possible mechanisms? The EU recommendation can also be considered. For a study like yours, using different concentrations of ZEA would be more appropriate, but this is just a suggestion for next time.
Further comments:
- 2.8: Statistical analysis: where was one-way analysis ANOVA used for three or more groups?

Round 2

Reviewer 2 Report (Previous Reviewer 2)

Dear, considering the manuscript submitted for review my opinion are as follows.

Thanks for your feedback, as well as answers to my doubts. As before, this is a very complex investigation, that requires researchers' scientific background to interpretations of results and study conclusions. I did not recognize it.
Zearalenone is one of the most spread mycotoxins and therefore preventive measures to reduce their presence in the feed and consequently, losses in production were well developed and implemented in the feed industry and on the farms level. Moreover, copper belongs to the essential microelements, mainly insufficient both in human and animal nutrition, and thus must be supplemented.
Therefore, the objectives and the rationale of the study are not clearly stated.
The authors not clearly emphasized the strengths of their study, or theory,  supported by physiological arguments.
I can not support such an investigation, due to unphysiological and practical implications. This investigation doesn't represent a realistic situation.   

Sincerely

Author Response

This manuscript is a resubmission of an earlier submission. The following is a list of the peer review reports and author responses from that submission.

Round 1

Reviewer 1 Report

Zearalenone (ZEA) is a prominent feed contaminant around the world and it is very harmful to livestock industry. Authors treated goat granulosa cells (GCs) with ZEA, it turned out that ZEA could promote the apoptosis and oxidative stress level of GCs, inhibit cell proliferation and arrest cell cycle. It was further found that copper transport related genes expressions were altered as well which leading to copper accumulation in GCs by using RNA-seq. But there are many issues, so I have decided that the manuscript is not acceptable in its present form and requires minor revision before it can be considered for publication.

1. Lack of reference citations when the results section states existing research results. Such as Line 199, "It has been demonstrated that ROS accumulation induced by mycotoxins can impair 199 cell viability and inhibit cell proliferation", authors should indicate the reference that reported the results. There are at least four errors like this.

2. Fig6A, Fig6B and Fig6C were not cited in the manuscript and if these pictures do not require in-depth explanation, please put in the supplementary material. In addition, the order of the pictures should be sorted according to the citation order in the manuscript, such as Fig4 and Fig5. The author should check all figures in detail and reorder the subfigures.

3. Please replace the figure of WB in Fig1C.

4. The coordinates of the histogram in the text should be changed to “relative expression level of mRNA/protein”.

5. Authors set 4 specified concentrations of ZEA (0, 50, 100 and 200 μmol/L) in Section 2.2 of Material Methods, but the following results section did not show the effects of the 4 treatments and why 200 μmol/L was used for subsequent experiments, please give corresponding figure to illustrate.

6. Please explain why the number of cells in S phase decreased while the number of cells in G2 phase increased after treatment with ZEA in Fig1A.

Reviewer 2 Report

This report describes the outcome of the relationship between ZEA exposure and Cu accumulation in GCs, and do ZEA might cause granulosa cell death through copper accumulation.

Despite high standards for the presentation of the results used, the manuscript lacks Novelty, because is not well-defined. Therefore the results do not provide an advancement of the current knowledge of International relevance. The negative effects of ZEA on the reproductive systems of the animals are well now investigated.

Also, in terms of Significance and Scientific Soundness, the manuscript had many shortcomings and flaws.

Also, in terms of Significance and Scientific Soundness, the manuscript had many shortcomings and flaws. My opinion is supported by the facts that the aim and scope of the study are not well linked with statements in the discussion and overall conclusion due to the authors did not able to directly prove that ZEA can impair GCs by the cuprotosis cell.

The overall conclusion is that the paper will not attract a wide readership, or be of interest only to a limited number of people.